# Structuring Annotation Label Spaces by Natural Language Concept Elicitation and Ontology Grounding

Pratik Sitapara[1,2,*], Prathmesh Doddanawar[1], Thiago S. Gouvêa[1,2] and Daniel Sonntag[1,2]

[1]*German Research Center for Artificial Intelligence (DFKI), Oldenburg, Germany*
[2]*University of Oldenburg, Germany*

## Abstract

Annotation remains a central bottleneck in data-centric AI workflows, particularly in expert-driven domains where labels encode complex and evolving domain knowledge. Existing annotation tools typically rely on flat, task-specific label sets that lack semantic structure, interoperability, and principled integration with established knowledge bases. Conversely, authoritative ontologies provide shared vocabularies and formal relations but are often too rigid and difficult for domain experts to navigate and extend during annotation setup. We present a demonstration system for LLM-assisted label space construction, enabling experts to define, structure, and ground annotation concepts through natural language interaction. The system follows a grounding-first strategy that prioritises alignment with authoritative ontologies and domain-relevant knowledge sources. When experts introduce novel concepts not covered by existing resources, the system generates local OWL-based concept representations with explicit provenance. The resulting session-specific ontology defines a fixed semantic contract for subsequent annotation, ensuring both semantic precision and task-level flexibility. We demonstrate the workflow in a bioacoustic annotation scenario and illustrate how human-in-the-loop, tool-mediated grounding can connect natural language concept elicitation with ontology-based label-space construction.(Video available at https://cst.dfki.de/projects-grounded-label-space-demo)

## Keywords

Grounded Label Space Engineering, Structured Knowledge, Knowledge Engineering, Knowledge-Centric Annotation

## 1. Introduction

The shift from model-centric toward data-centric AI has made data annotation a central bottleneck in the development of robust and reliable learning systems, particularly in expert-driven domains where labels encode domain knowledge rather than merely task-specific class names [1, 2, 3]. In such settings, annotation is not merely a process of assigning predefined labels, but also a process of meaning construction in which experts interpret observations, refine distinctions, and negotiate category boundaries [1, 4].

However, most existing annotation workflows rely on label spaces that are *flat*, *fixed*, and *ungrounded*, offering limited support for structured semantics, evolving concepts, and principled linkage to external knowledge resources [5, 6]. This limitation is particularly critical because knowledge bases and ontologies provide shared vocabularies, hierarchical structure, and explicit semantic relations that enable abstraction and generalisation across labels, principled handling of granularity, interoperability across datasets, and consistency checking through validation and reasoning [7, 8, 9]. Without such grounding, annotation decisions remain local and implicit, reducing their interpretability and limiting their downstream utility across datasets and tasks.

This limitation is reflected in both mainstream annotation tools and existing semantic annotation

*GenAIK-NORA 2026: Joint Workshop on Generative AI and Knowledge Graphs and KNOwledge GRaphs & Agentic Systems Interplay, co-located with IJCAI-ECAI 2026, August 17, 2026, Bremen, Germany*

*Corresponding author.

✉ pratik_popatbhai.sitapara@dfki.de (P. Sitapara); prathmesh.prakash_doddanawar@dfki.de (P. Doddanawar); thiago.gouvea@dfki.de (T. S. Gouvêa); daniel.sonntag@dfki.de (D. Sonntag)
🌐 https://github.com/yapat-app/oe_yapat (P. Sitapara)
🆔 0009-0001-2456-4586 (P. Sitapara); 0009-0006-8164-2647 (P. Doddanawar); 0000-0002-0727-5838 (T. S. Gouvêa); 0000-0002-8857-8709 (D. Sonntag)

platforms. Widely used systems such as CVAT [10], Label Studio [11], and Prodigy[12] are effective for scalable data labelling, but they generally assume a task-specific schema that is defined in advance, are flat lists or lightly structured taxonomies [13, 14, 6]. More knowledge-aware platforms, such as INCEpTION [15], support interactive and semantic annotation, including concept linking and knowledge-base-assisted annotation. In parallel, ontology engineering environments such as WebProtégé [16] and VocBench [17] provide mature support for collaborative ontology development. However, these tools address different stages of the workflow: annotation systems primarily support labelling under an existing schema, whereas ontology editors presuppose a dedicated ontology engineering setting. Consequently, they provide limited support for the interactive construction of an ontology-informed label space by domain experts *prior* to annotation.

The problem becomes more acute in expert domains characterised by conceptual ambiguity, domain shift [18], and ongoing knowledge production. Flat label spaces collapse hierarchy and semantic relations into local identifiers, which limits abstraction, reuse, and interoperability across datasets and tasks [19, 20]. Fixed schemas further assume that all relevant categories are known in advance, an assumption that breaks down when experts encounter novel phenomena, outdated classifications, or changing domain distinctions [18, 21]. Ungrounded labels compound this issue because their meaning often remains implicit in local guidelines or annotator conventions rather than being explicitly linked to shared ontologies or knowledge graphs [7, 22]. Although recent LLM-based work has shown promise for ontology generation [23, 24], alignment [25], population [26], and conversational ontology engineering [27, 28], including systems such as OntoChat [29], Konda [30], and OntoGPT [31] these approaches mainly target offline ontology construction or post hoc semantic processing rather than supporting interactive, pre-annotation label-space design.

In this paper, we present an interactive demonstration system for *grounding-first label-space construction* prior to annotation in a wildlife acoustic monitoring setting [32, 33, 34]. Before annotation begins, domain experts define relevant concepts through natural language interaction. The system prioritises grounding in authoritative ontologies and domain-relevant resources, and when no suitable external match exists, it generates local OWL-based ontology modules with explicit provenance metadata. The resulting session-specific ontology serves as a semantic contract for subsequent annotation, combining semantic precision with task-level flexibility while preserving transparency and reproducibility [35].

## 2. System Description

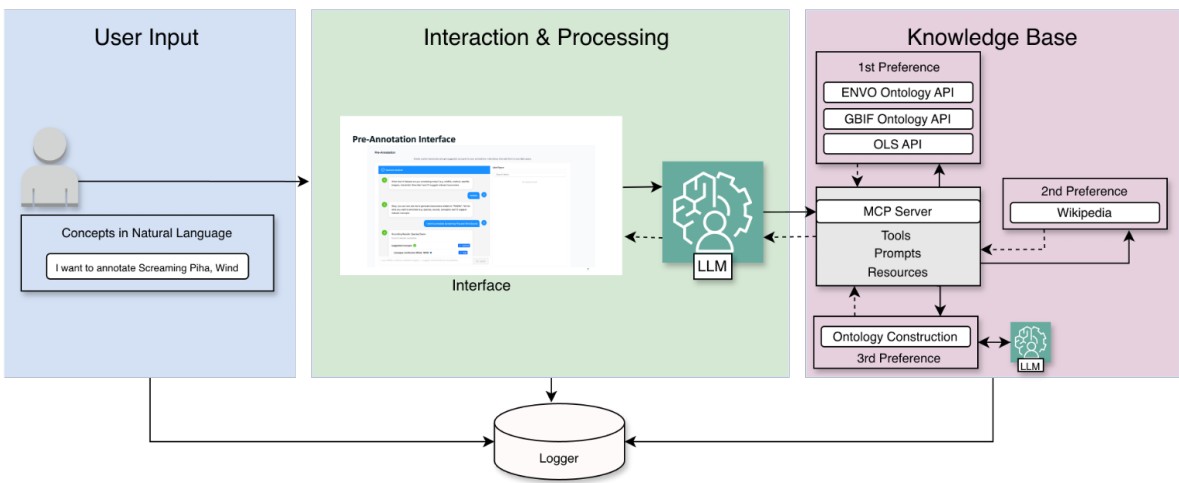

**Figure 1: System architecture**. Experts provide concepts in natural language; an LLM, mediated by an MCP server, invokes ontology and knowledge base APIs (prioritising authoritative sources) to retrieve candidates. Confirmed concepts are incorporated into a session ontology, and all grounding decisions are logged for provenance and reproducibility.

The system implements a framework for ontology-grounded label space construction through natural language interaction. Its objective is to couple expert concept elicitation with explicit ontology engineering primitives under controlled, inspectable execution. Semantic structuring is performed prior to annotation, establishing a transparent, auditable, and reproducible semantic contract that governs subsequent labelling. The current instantiation is demonstrated in a bioacoustic annotation setting with prioritised grounding to domain-relevant ontologies.

## 2.1. Design Principles

The system is guided by three architectural principles:

- **Grounding-First Semantics** Every concept is expected to align with authoritative ontologies when possible. Concepts that cannot be grounded remain explicitly marked as local rather than implicitly absorbed into informal label sets.
- **Structural Separation** Ontology construction is completed prior to annotation, establishing a stable semantic contract that constrains subsequent labelling.
- **Mediated Control** Natural language interaction does not directly modify ontology state; all structural changes require explicit, tool-mediated execution and user confirmation.

## 2.2. Architecture Overview

Figure 1 presents the system architecture, organised into three logically decoupled layers that separate user interaction, LLM-based orchestration, and ontology operations. The interaction layer is implemented as a React frontend built with the Vite framework and TypeScript, enabling structured session management and typed communication across components. This layer maintains the session state, including candidate concepts, confirmed labels, grounding status, and provenance metadata. It supports a two-phase workflow consisting of a discovery phase, in which concepts are iteratively elicited and grounded, and a frozen phase, in which the resulting label space is fixed and exported as a session ontology.

The system core is implemented as a Python-based backend and orchestration layer driven by an OpenAI language model (gpt-4o [36]). This layer manages the interaction workflow, interprets natural language input, extracts candidate concepts, and generates structured tool invocations. Ontology-related operations are not executed directly by the language model. Instead, all ontology-related operations are executed through the MCP server, which exposes typed, inspectable tools for lookup, alignment, local ontology construction, and validation. Tool outputs are returned as grounding candidates. Only upon explicit user confirmation are selected candidates incorporated into the session ontology maintained by the client. This ensures that semantic state changes are mediated, logged, and reproducible. This separation increases transparency, reduces the risk of uncontrolled semantic drift, and ensures that semantic state changes remain auditable and reproducible.

Accordingly, the system distinguishes between two concept categories within the session ontology: authoritatively grounded concepts, which are linked to confirmed entities in external ontologies, and locally defined concepts, which are represented in OWL with structured semantics and provenance but are not yet aligned to external ontology entities. All interactions between the expert, language model, and MCP server are logged in a structured format, including tool invocations, retrieved candidates, and user confirmations. This logging mechanism provides a complete provenance trace of the label-space construction process. Overall, the architecture enforces a strict separation between natural language interactions and ontology tasks while enabling controlled, human-in-the-loop construction of semantically structured label spaces.

Comprehensive supplemental materials, including a step-by-step workflow demonstration with grounding status categories, a user interface, and an example of a structured representation of a novel concept, are provided in Appendix 4.

# 3. Discussion

This work introduces grounding-first label space construction as an explicit pre-annotation workflow that links interactive annotation with ontology engineering under human control. Instead of treating semantic alignment as a post hoc step, the approach performs semantic structuring before annotation and produces a session ontology that serves as a *semantic contract* for subsequent labelling.

This design addresses a central limitation of conventional annotation workflows. Flat label spaces are efficient, but collapse hierarchical structure and semantic relations, while post hoc grounding reconstructs meaning only after annotation decisions have already been made [5, 37]. By making semantic commitments explicit during schema creation, the proposed workflow improves interoperability while still allowing experts to introduce locally meaningful concepts through controlled ontology modules.

Beyond improved schema clarity, structured label spaces enable capabilities that are not accessible in conventional workflows. In particular, the resulting ontological representation supports hierarchical abstraction, multi-granularity annotation, and consistency checking through reasoning. Moreover, explicit grounding facilitates interoperability with external knowledge bases and enables reuse across datasets, reducing duplication of conceptual modelling effort. The inclusion of locally defined concepts, together with provenance metadata, further enables the capture of emerging domain knowledge without sacrificing transparency or future integrability.

Overall, the proposed approach demonstrates how annotation can be reframed as a knowledge-centric process, in which label spaces are not static inputs but structured, evolving artefacts. This perspective provides a concrete step toward integrating data annotation with formal knowledge representation, supporting more interpretable, adaptable, and interoperable AI systems.

# 4. Limitations and Future Work

This work presents a system and workflow demonstration rather than a full empirical study. While the proposed architecture enables controlled and reproducible label space construction, several limitations remain. First, locally introduced novel concepts are currently preserved as structured, session-specific representations but are not yet systematically aligned with entities in existing ontologies or knowledge bases. Enabling candidate retrieval, mapping, and expert-validated alignment will be important for improving interoperability and facilitating consolidation into shared semantic resources. Second, ontology coverage is limited to a small set of domain resources; extending support to additional standards (e.g., Darwin Core, NCBI Taxonomy, PATO) is necessary to broaden applicability.

A central open question concerns on evaluating grounding-first label space construction relative to existing annotation paradigms. We propose a comparative evaluation along three axes, contrasting (i) conventional flat-label annotation, (ii) annotation with fixed ontology-derived label spaces, and (iii) the proposed grounding-first approach. Evaluation should distinguish between workflow-level and knowledge-level outcomes.

At the workflow level, user studies with domain experts can assess annotation efficiency, cognitive load, and usability, as well as inter-annotator agreement under evolving or ambiguous concepts. At the knowledge level, the resulting label spaces can be assessed for grounding quality, semantic consistency, provenance completeness, and reuse across datasets or sessions. At the downstream level, models trained on annotations from each condition can be compared under domain shift to test whether semantically structured label spaces improve robustness and transfer. Beyond quantitative evaluation, collaboration with ontology engineers will be essential to assess whether locally generated modules can be integrated into existing ontologies and whether the captured provenance supports long-term knowledge curation. Together, these evaluation dimensions reflect the dual objective of the approach: improving both the annotation workflow and the quality of the resulting semantic artefacts. Future work will therefore focus on implementing this evaluation protocol and strengthening the validation of LLM-generated ontology modules.

## Acknowledgments

This research is part of the Computational Sustainability & Technology project area[1], and has been supported by the Lower Saxony Ministry of Science and Culture (MWK) in zukunft.niedersachsen program, the Federal Ministry of Research, Technology and Space (BMFTR) under grant number 16IW26002 (AMENABLE), and the Endowed Chair of AAI at the University of Oldenburg.

## Declaration on Generative AI

During the preparation of this work, the author(s) used AI tools to perform grammar and spelling checks. LLMs are also part of the described system. After using these tool(s)/service(s), the author(s) reviewed and edited the content as needed and take(s) full responsibility for the publication's content.

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

## A. Workflow

We demonstrate the proposed workflow in a bioacoustic annotation setting, illustrating how natural language concept elicitation, ontology grounding, and local ontology construction are carried out prior to the labelling phase. The focus is on the structured assembly of a session-specific ontology that will serve as a fixed semantic contract during annotation.

### A.1. Application Scenario

Dr. Aris, an ornithologist specialising in neotropical bird communities, prepares to annotate a corpus of passive acoustic monitoring (PAM) recordings [32, 38]. The task requires distinguishing focal bird species, background taxa, and abiotic environmental sounds (e.g., rain, wind). While Dr. Aris is a domain expert, he does not routinely work with ontology identifiers, IRIs, or formal schema structures from resources such as GBIF or ENVO. The system therefore supports him in expressing relevant distinctions in natural language while transparently grounding concepts in authoritative ontologies wherever possible.

**Step 1: Natural Language Concept Elicitation.** Dr. Aris is greeted in the pre-annotation interface and replies with a request:

> **System:** I am configured for audio data in the domain of bioacoustics. This session will define the label space for sound event detection. What target sound categories do you intend to include in your annotation schema?
> **Dr. Aris:** I want to annotate *Poecile montanus*, *Screaming Piha*, and wind sounds.

The system interprets this input as candidate ontology concepts rather than flat labels.

**Step 2: Grounding via Tool Invocation.** The assistant parses the input into distinct concepts and invokes ontology lookup services through the MCP server. Taxonomic entities are queried via GBIF, while environmental phenomena are queried via ENVO. No identifiers are incorporated into the session ontology without tool confirmation.

**Step 3: Retrieval and Candidate Presentation.** The MCP server retrieves candidate classes including stable identifiers, labels, and definitions. For example, taxonomic records are returned for *Poecile montanus* and *Lipaugus vociferans* (Screaming Piha), and environmental ontology entries such as "atmospheric wind" are retrieved for wind. The candidates are presented to Dr. Aris for review.

**Step 4: Human-in-the-Loop Grounding.** Dr. Aris selects the appropriate options among the offered candidates for each concept. Upon confirmation, each concept is grounded, and the confirmed grounding source is recorded and logged.

**Step 5: Finalisation.** After confirming all concepts, Dr. Aris freezes the pre-annotation stage. The confirmed labels are consolidated into a session-specific ontology that defines the structured label space for annotation. The ontology can be exported in standard semantic formats (e.g., OWL or JSON-LD), preserving grounding status and provenance metadata. Annotation proceeds using this fixed ontology.

## A.2. Grounding Status Categories

Each confirmed concept in the session ontology is assigned an explicit grounding status based on its source.

**Authoritative Ontology Grounding** When a concept can be matched to a class in a domain-relevant ontology (e.g., GBIF for biological taxa or ENVO for environmental phenomena), the corresponding identifier is retrieved via MCP tools and, upon confirmation, incorporated into the session ontology.

**Knowledge-Base Grounding** If no suitable match is found in authoritative ontologies but a corresponding entity exists in a general-purpose knowledge base (e.g., Wikidata, DBpedia), the concept may be linked to that resource. The confirmed identifier is stored together with its grounding source.

**Local Concept (Ungrounded)** In cases where a concept is not present in existing ontologies or knowledge bases, the system performs modular ontology construction. The expert provides a natural language description including observable properties (e.g., morphology, behaviour, habitat). Using LLM-assisted modelling, the system generates a local modular ontology module representing the concept. Such concepts are explicitly marked as locally defined and retain the structured description extracted from the expert input.

# B. User Interface

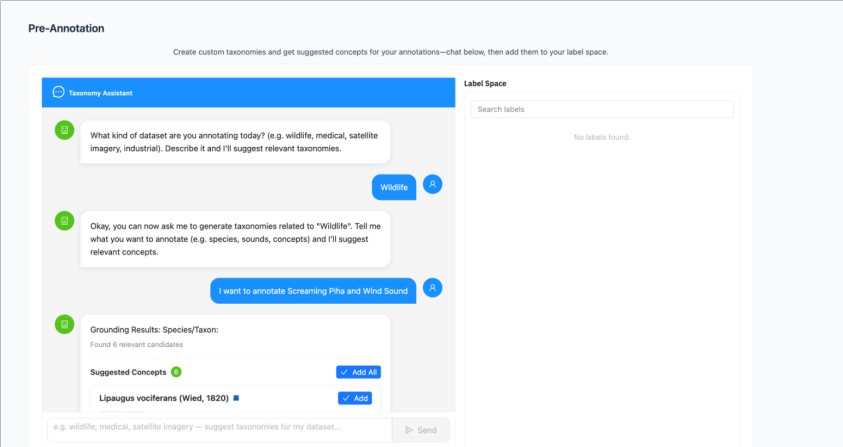

**Figure 2: Pre-annotation interface**. The interface supports natural-language concept elicitation and ontology-assisted label-space construction prior to annotation.

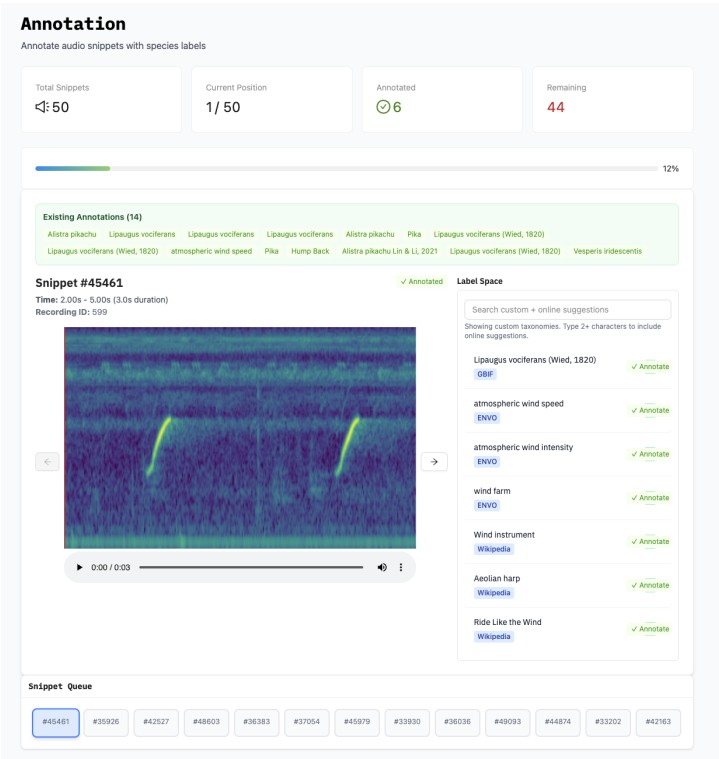

**Figure 3: Annotation interface**. After the pre-annotation phase is finalised, the confirmed label space is used during annotation. The interface presents audio snippets together with spectrograms, existing annotations, and ontology-informed label suggestions for controlled and consistent semantic labelling.

## C. Local concept representations

The following listing illustrates the structured representation of a novel species, *Vesperis iridescentis*, as a populated OWL ontology module serialised in Turtle syntax. This formal output was generated by mapping expert-provided natural-language descriptions into a modular knowledge schema that integrates morphological, ecological, and behavioural dimensions.

Listing 1: Populated OWL ontology module for *Vesperis iridescentis* (Shimmer-Veil Moth), generated from expert local concepts.), generated from expert local concept in natural language.

```
@prefix beh: <http://yourproject.org/ontology/behavior#> .
@prefix eco: <http://yourproject.org/ontology/ecology#> .
@prefix morph: <http://yourproject.org/ontology/morphology#> .
@prefix owl: <http://www.w3.org/2002/07/owl#> .
@prefix rdfs: <http://www.w3.org/2000/01/rdf-schema#> .
@prefix tax: <http://yourproject.org/ontology/taxonomy#> .

<http://yourproject.org/data/species_vesperis_iridescentis> a owl:Ontology ;
    rdfs:label "Ontology Module for Vesperis iridescentis" ;
    owl:imports <http://yourproject.org/ontology/core> .

tax:tc_vesperis_iridescentis a tax:TaxonConcept ;
    rdfs:label "Vesperis iridescentis" ;
    beh:hasBehavior beh:vesperis_iridescentis_activity_period,
        beh:vesperis_iridescentis_defensive_action,
        beh:vesperis_iridescentis_flight_pattern ;
    eco:hasHabitat eco:habitat_mistladen_highaltitude ;
    eco:interactsWith tax:tc_highaltitude_flycatcher,
        tax:tc_lunaorchid ;
    morph:hasTrait morph:vesperis_iridescentis_antennae_length,
        morph:vesperis_iridescentis_body_color,
        morph:vesperis_iridescentis_wing_color,
        morph:vesperis_iridescentis_wing_texture,
        morph:vesperis_iridescentis_wingspan ;
    tax:commonName "Shimmer-Veil Moth".
```

## D. Online Resources

The source code and demonstration video for the presented tool are available at:

- GitHub,
- Video.