# OpenReview forum: "Structuring Annotation Label Spaces by Natural Language Concept Elicitation and Ontology Grounding"
_ijcai.org/IJCAI-ECAI/2026/Workshop/GENAIK-NORA — IJCAI-ECAI 2026 Joint Workshop on GENAIK and NORA_

### Official Review · Reviewer_BFfQ · 2026-05-31
**paper presents a case study of a developed tool without evaluation of the impact**

**Rating:** 6
**Confidence:** 4

**Review:**

One of the main challenges in data annotation is defining annotation guidelines that are critical to achieve high inter annotator agreement.
The authors present present a tool that enables annotators to ground the annotation schema to either existing ontologies or wikipedia.
The proposed approach also allows to extend the annotation schema during the annotation. No evaluation is provided of the impact of the created tool.
The presented paper strikes the perfect balance between Knowledge Graphs, Generative AI & Agentic Systems Interplay for a demo with a case study.

**Novelty:**
The authors introduce the idea of creating a conversational interface to ground flat text labels to existing ontologies. This is a new application of conversational AI combined with a mcp tool that gives access to publicly available ontologies for bio-acoustic annotation.

**Clarity:**
While it's clear what the authors developed, this work presents only a system and workflow presentation without any evaluation. It's not clear how widely applicable is this approach. Also it's not clear how allowing the expert to introduce new labels during the annotation impacts the task.

**Reproducibility:**
The authors provide the code github link.

**Ethical Compliance:**
No ethical concerns, societal impacts, or biases.

**Open Questions:**
- How often annotation task can be grounded on existing on ontologies?
- Does the defined approach help in simplifying annotation guidelines?
- Does the defined approach shorten the time to define annotation guidelines?
- Does the described approach increase inter annotator agreement?
- What happens when multiple experts use the described tool?

---

### Official Review · Reviewer_5WQT · 2026-06-02
**The system design is promising, but the paper lacks strong rationale or empirical support**

**Rating:** 6
**Confidence:** 4

**Review:**

**Summary** This demo paper presents an LLM-powered system for constructing annotation label spaces through concept grounding and generation from natural language interaction. The system follows a grounding-first strategy that prioritizes alignment with authoritative ontologies and domain-relevant knowledge sources via MCP tools, rather than relying solely on the LLM for concept generation. These grounded concepts inherently have rich internal structure derived from the original sources. When experts introduce novel concepts not covered by existing resources, the system generates local concept representations with explicit provenance. These grounded and generated concepts are organized into a session-specific ontology for the target annotation task. This label space construction is performed separately from annotation to provide a fixed, stable semantic basis for subsequent annotation. The paper discusses these design choices and their motivations, and illustrates the workflow with a use case in bioacoustic annotation.

Overall, the system is well designed and implemented, but I am not fully convinced by some of the claimed advantages of the proposed approach. The paper would benefit from stronger rationale and/or empirical evidence.

**Pros**

1. The system is thoroughly implemented, with a clear UI and rich data support.
2. The paper is well written and clearly describes its underlying design choices and how the system works.
3. The key design choices are largely reasonable (although I am not fully convinced by the claimed advantage of separating ontology construction from annotation). In particular, the grounding-first strategy is an interesting and reasonable approach that could stimulate future work in this direction.

**Cons**

1. The system assumes upfront label space construction, which requires highly competent and reliable human users who can articulate the target concepts and monitor their quality before starting annotation. In my experience, data annotation is almost always an iterative process in which the label space is refined as annotators see more data and encounter edge cases. So the proposed separation between ontology construction and annotation seems challenging to achieve in practice. (That being said, I think the system could also be used in an iterative way. I wonder why the authors strictly aim to finalize the label space before annotation.)
2. The paper would be more convincing if it provided empirical evidence that this approach leads to better outcomes than traditional iterative annotation processes. For example, a user study could help identify what works well and what is still missing. Currently, the paper does not provide many insights into real use cases or potential future directions.
3. The paper claims improvement over flat label spaces, but this advantage is not fully realized because the grounded concepts and generated concepts are not systematically aligned with each other. Similarly, the paper criticizes fixed label spaces for their weak adaptability, but the proposed system also fixes the label space before annotation without considering possible changes afterward. After reading the rest of the paper, the introduction sounds somewhat over-promising and slightly misleading.

**Comments**

1. The text in Figure 1 is too small to read.
2. In the third line of the second paragraph in Section 1: Knowledge bases -> knowledge bases
3. While the codebase is available, it does not have any documentation or instructions for using the system. As a demo paper, it would be more helpful if the codebase could be made more accessible for readers who want to try out the system.

---

### Official Review · Reviewer_vKwV · 2026-06-05
**A principaled approach to leveraging LLMs for the development of annotation labels**

**Rating:** 10
**Confidence:** 4

**Review:**

Annotation is an important part of developing datasets, especially for smaller and more specialized domains. It is sometimes not treated with sufficient attention, especially the initial steps of annotation, which involve developing annotation labels. This paper describes a process for leveraging existing ontologies that guides expert annotators in adding annotation labels to data through a natural language conversation. The paper presents a user interface that is suitable for use by domain experts who may not be familiar with ontology development tools. The system assists the user in mapping concepts to ontologies and in generating new concepts if needed. It labels new concepts as "local" to prevent them from being added to an ontology permanently without review.

This is a very interesting paper and opens up some new directions in annotation that could lead to much improved annotation, better datasets, and finally, more accurate classification systems. One suggestion I have is that the paper would be improved by some discussion of how this system would scale to a larger number of annotators. If a number of annotators are at work at one time, their decisions could easily become inconsistent. How would such inconsistencies be reconciled? Perhaps the system could include some kind of central database or ontology so that different annotators could see each other's work in progress.

It might also be worth discussing what happens when the efforts start with different sizes of ontologies. What is the minimum ontology size for this process to be effective?

The authors might also consider reviewing efforts like the Penn Treebank annotation label development process, which was a very well-thought-out effort involving multiple annotators but not starting from an ontology.

Finally, it is interesting to see a combination of LLMs with knowledge-based components that leverages the strengths of each technoloy and should be of great interest to the attendees at this workshop.

---

### Official Review · Reviewer_34NC · 2026-06-07
**Interesting demo paper describing a tool for semantic annotation, ontology engineering, and human-in-the-loop generative AI workflows for knowledge-grounded AI systems,**

**Rating:** 8
**Confidence:** 4

**Review:**

The paper presents an annotation tool that supports ontology-grounded annotation using human-in-the-loop and leveraging an LLM. The proposed approach introduces a grounding-first workflow in which domain experts define annotation concepts before annotation begins. Concepts are aligned with authoritative ontologies when possible, while novel concepts are represented as local OWL modules with provenance information. The paper addresses a genuine problem in annotation workflows: the lack of semantic structure, interoperability, and ontology grounding in existing annotation platforms. The work sits at the intersection of generative AI, ontology engineering, knowledge graphs, and human-in-the-loop annotation, making it relevant to the workshop themes.

The system is demonstrated in a bioacoustic annotation scenario.

This is a solid demonstration paper that addresses an important problem in semantic annotation and ontology-aware AI workflows.

Strengths:
- Well-motivated work. The paper provides evidence that their work fills an existing gap.
- Timely and interesting paper. Annotation tools are important for the development of datasets.
- The work addresses an important problem or gap in structured semantic annotation.
- The architecture is well designed.
- The paper is generally well written and easy to follow.

Weaknesses:
- The primary limitation is the absence of empirical validation already acknowledged by the author(s).
- The notion of a "semantic contract" is attractive and appears throughout the paper. However, it is not formally defined, and the paper offers no examples of reasoning or validation. Please elaborate on the notion further.
- Repetitions throughout the paper.

---

### Decision · Program_Chairs · 2026-06-10

Accept